# Automated Real-Time Evaluation of Condylar Movement in Relation to Three-Dimensional Craniofacial and Temporomandibular Morphometry in Patients with Facial Asymmetry

**DOI:** 10.3390/s21082591

**Published:** 2021-04-07

**Authors:** Won-June Lee, Ki-Ho Park, Yoon-Goo Kang, Su-Jung Kim

**Affiliations:** Department of Orthodontics, Kyung Hee University School of Dentistry, Seoul 02447, Korea; knight8610@naver.com (W.-J.L.); pkhmate@hanmail.net (K.-H.P.); deodor@hanmail.net (Y.-G.K.)

**Keywords:** mandibular asymmetry, temporomandibular joint, condylar movement, cone-beam computed tomography, ultrasonic jaw-tracking system

## Abstract

The aim of this study was to investigate the correlation between craniofacial morphology, temporomandibular joint (TMJ) characteristics, and condylar functional movement in patients with facial asymmetry using an up-to-date automated real-time jaw-tracking system. A total of 30 patients with mandibular asymmetry and prognathism were included. Three-dimensional (3D) craniofacial and TMJ morphometric variables were analyzed in images captured using cone-beam computed tomography. Three-dimensional condylar movements were recorded during the opening, protrusion, and laterotrusion of the jaw and divided into those for deviated and non-deviated sides. Overall functional and morphometric variables were compared between the sides by a paired *t*-test. Pearson’s correlation analysis and factor analysis were also performed. As a result, significant differences were found between the sides in morphometric and functional variables. The condylar path length was significantly longer and steeper on the deviated side during protrusion and lateral excursion. TMJ morphometric asymmetry, more so than the craniofacial morphologic asymmetry, seemed to be reflected in the functional asymmetry, representing different correlations between the sides, as supported by factor analysis. This study provides evidence explaining why the asymmetric condylar path remained unchanged even after orthognathic surgery for the correction of craniofacial asymmetry.

## 1. Introduction

Facial asymmetry encompasses mandibular functional asymmetry as well as craniofacial morphologic asymmetry. The amount and direction of mandibular deviation in patients with facial asymmetry may be misestimated in the presence of mandibular functional asymmetry [1,2,3,4,5]; thus, incorrect diagnosis and treatment planning can result due to the hidden discrepancies remaining undetected. Moreover, it was speculated that there is a relationship between morphologic and functional asymmetries in the temporomandibular joints (TMJs) and, accordingly, facial asymmetry might be a causative factor of temporomandibular disorders (TMD) [6]. Considering that the functional rehabilitation of TMJs should be one of the main goals of orthodontic and orthognathic treatments, dynamic functional evaluation should be interpreted in accordance with morphologic examination, particularly in patients with facial asymmetry who require orthognathic surgery.

A jaw-tracking system for dynamic functional analysis of mandibular movement has made rapid progress along with the development of electronic recording instruments [7]. Three representative real-time recording systems that are currently utilized are the opto-electric [6,8,9], electromagnetic [10,11], and ultrasonic systems [12]. As a computerized ultrasonic axiography, the AxioQuick^®^ recorder (SAM Co., Munich, Germany) is specialized in quantitative analysis of the direction and amount of condylar paths within the glenoid fossa during mandibular border movement [13,14,15]. This registration system is based on the measurement of real-time latency periods of sequentially transmitted ultrasound pulses between four transmitters attached to the mandible and eight receivers mounted on a face bow [14]. The improved resolution of the 3D sensors increased the signal quality and diagnostic validity. Inclusion of lightweight tiny sensors could enhance the patient’s comfort during jaw movement and thus decrease the measurement errors. High diagnostic specificity and sensitivity of this system has been proven in both children [13] and adults with healthy or pathologic TMJs [14,15]. Based on the reliability and validity on the clinical relevance, a superior AxioQuick^®^ recorder system was introduced in the present study to investigate the diagnostic value of this to analyze various condylar movements in the specific condition of dentofacial asymmetry.

As for the relationship between condylar movement and TMJ morphologic characteristics in patients with facial asymmetry, the asymmetrical condylar position and differences in path lengths between the deviated side (DS) and the non-deviated side (NDS) have been noticed in previous studies [16,17,18,19]. The condyles at DS tended to be positioned more posterosuperiorly with increasing mandibular deviation, and the condylar path length at DS was significantly longer than that at NDS. In addition, both the sagittal condylar path angle and anterior wall of the glenoid fossa were steeper at the DS than at the NDS [20,21]. Based on the finding that morphologic asymmetry is reflected in functional asymmetry, it was postulated that the condylar path tends to compensate for the morphologic asymmetry during jaw movement [6]. However, most previous studies have relied on two-dimensional posteroanterior cephalometric analysis, and could thereby not explain certain variations in the mandibular movement in relation to the variations in TMJ morphology according to the different types of craniofacial asymmetry.

There have been several studies on the relationship between facial morphology and mandibular movement [9,22]. Mouth opening capacity was found to be positively correlated with mandibular length and negatively correlated with ramal inclination, sagittal jaw relation, and mandibular plane steepness. On the other hand, Ikeda et al. [23] demonstrated that in the group with facial asymmetry, there were significant correlations between the asymmetric ratios of condylar path length and inclination and mandibular morphology, whereas there were no significant correlations between these factors in the control group with a menton deviation of less than 4 mm. Based on the premise that three-dimensional (3D) condylar paths may be affected by bilaterally different conditions of mandibular morphology in patients with facial asymmetry, we sought to investigate if functional asymmetry could be predicted in different types of morphologic asymmetry when considering DS and NDS separately.

The purpose of this study was to elucidate the craniofacial and TMJ morphometric features affecting asymmetric condylar movement between the DS and NDS in patients with facial asymmetry based on a comparison of overall functional and morphometric variables between the sides when using cone-beam computed tomography (CBCT) analysis and a 3D automated real-time jaw-tracking system.

## 2. Materials and Methods

### 2.1. Subjects

The study protocol was approved by the Institutional Review Board of Kyung Hee University Dental Hospital, Seoul, Korea (KHD IRB 1612-3). Forty three patients with facial asymmetry and mandibular prognathism were tested using an ultrasonic jaw-tracking system before orthodontic treatment, from January 2017 to December 2019. We finally evaluated 30 patients with a mean age of 21.5 ± 2.3 years (ranging 18.2–27.5 years). The inclusion criteria were as follows: (1) facial asymmetry, defined as a chin deviation greater than 3 mm from the facial midline at the maximum intercuspal position; (2) mandibular prognathism, defined as chin protrusion (distance from pogonion to nasion perpendicular vertical line >0 mm, ANB (A point-nasion-B point angle) < 0°); (3) high quality CBCT images and functional records. The exclusion criteria were (1) subjective TMJ pain and motion limitation; (2) pathologic condylar resorption or deformation in the CBCT images; (3) craniofacial anomalies and syndrome; (4) history of facial trauma or pathologic jaw bone disease; (5) systemic diseases or medication.

### 2.2. 3D Morphometric Analysis

The CBCT scan was performed before treatment (PSR 9000N, Asahi Roentgen, Kyoto, Japan; 80 kvp, 10 mA, 30-s scan time, 0.1 mm^3^ voxel size) to analyze 3D craniofacial and TMJ morphologies. The data were reconstructed as 3D images using InVivo^®^ Dental 5.3 Software (Anatomage, San Jose, CA, USA). One experienced examiner (LWJ) performed the CBCT image reconstruction and measurements two times in a 2 week-interval. The averaged values of the two data sets were taken for the analysis. The method error for each parameter was calculated using Dahlberg’s formula. The measurement errors ranged from 0.02 to 0.16 mm for the linear parameters and from 0.04 to 0.31 degrees for the angular parameters, indicating high intra-examiner reliability. Three-dimensional morphometric analysis was conducted in two aspects: craniofacial and TMJ morphologies.

#### 2.2.1. Craniofacial Morphology

Facial asymmetry was defined by the distance of menton deviation from the midsagittal reference plane. The Frankfurt horizontal (FH) plane was constructed by both sides of porion and left orbitale, and the midsagittal reference (MSR) plane was perpendicular to the FH plane passing through the nasion and basion point. For the evaluation of craniofacial asymmetric pattern, 10 landmarks were identified, and 5 parameters were measured on the CBCT images (Figure 1): maxillary height (MxH), the shortest distance from the FH plane to the central fossa of the maxillary first molar; ramal height (RH), the distance between the highest point of the condyle and the gonion; frontal ramal inclination (FRI), the angle formed by the FH plane and the lateral border of the ramus in the frontal view; lateral ramal inclination (LRI), the angle formed by the FH plane and the posterior border of the ramus in the sagittal view; and mandibular body length (BL), the distance between the menton and the gonion in axial view.

#### 2.2.2. Temporomandibular Joint (TMJ) Morphology

TMJ morphometric parameters consisted of three aspects: condylar position relative to the cranial base, condylar position relative to the glenoid fossa, and the shape of articular eminence (Figure 2). For the 3D evaluation of the condylar position, the coronal reference plane, which was perpendicular to the FH and MSR plane passing through the basion point, was established. On the axial image at the level of the line connecting the medial and lateral poles of the condylar head, the intersection of two lines passing through the largest lateromedial width and anteroposterior width of the condyles was determined as the center of the condyle point. For the evaluation of the condylar position relative to the cranial base, three linear parameters were defined as follows: anteroposterior condylar posture (APCP), transverse condylar posture (TCP), the shortest distance from the center of the condyle point to the midsagittal reference plane in axial view; vertical condylar posture (VCP), the shortest distance from the center of the condyle point to the FH plane in coronal view. For the assessment of condylar position within the glenoid fossa, five linear and one angular parameters were identified: superior joint space (SJS), anterior joint space (AJS), posterior joint space (PSJ), medial joint space (MJS), lateral joint space (LJS), and axial condylar angle (ACA). The reference lines to measure the joint spaces included a horizontal reference line parallel to the FH plane and tangent to the highest point of the superior wall of the glenoid fossa, and other lines tangent to the most prominent points of the condyle anteroposteriorly and mediolaterally (Figure 2A,B). For the measurement of the steepness of articular eminence, four angular parameters were defined—the anterior eminence steepness (AES), posterior eminence steepness (PES), lateral eminence steepness (LES), and medial eminence steepness (MES)—which were measured based on the best-fit line method (Figure 2C,D) [24,25].

### 2.3. 3D Mandibular Movement Analysis

Mandibular border movement was recorded using a computerized real-time AxioQuick^®^ recorder as an ultrasonic jaw-tracking system (Figure 3). All measurements were performed in an isolated room equipped with this recording system, where the room temperature is maintained without noise. Each patient was seated in an inclined supine position with the full device sets on the head and the mandible. Each patient was instructed to practice and repeat all tested mandibular movements until representative real-time estimates of all dynamic parameters could be obtained. At that time point, two measurement data sets were acquired in each patient. The averaged value of each parameter was taken for the analysis.

Geometric analysis of jaw movements was conducted by a computer-assisted mapping software program (AxioQuick^®^ recorder software, version 0.0.65, SAM Co., Munich, Germany) (Figure 4). Axis-horizontal plane was established as a zero-reference plane for all 3D mandibular movements. The patient’s terminal hinge axis was determined as the condylar reference point by rotational analysis in the software program automatically. The condylar paths were recorded during the mandibular movements in 3D: maximum opening and closing, protrusion, and working and non-working lateral movements. Along with 3D acquisition of movement, the software generated real-time digital data. The X, Y, and Z axes-based coordinated information was obtained for automatic quantification of seven dynamic parameters: opening condylar path length during maximum open–close movement (OCPL); protrusive condylar path length (PCPL); sagittal condylar inclination during protrusive movement (SCI); transverse condylar inclination during protrusive movement (TCI); non-working sagittal condylar path length (NCPL) at the balancing side; non-working incisal path length (NIPL) at the midpoint of the lower central incisors; and the Bennett angle (BA) measured at 1 mm from starting point.

### 2.4. Statistical Analysis

Power analysis was performed to determine the sample size needed for comparing the parameters between the DS and NDS with a 0.05 two-sided significance level. Based on that, the sample size required for 80% power for the significance levels of representative CBCT parameters was 20 (Table 1), and 30 subjects were finally analyzed in the present study.

Following the Shapiro–Wilk test to assess the normality of data distribution, a paired *t*-test was performed to compare between DS and NDS for each parameter. Out of the total 23 variables, variables that significantly differed between the sides as revealed by the paired *t*-test were selected for the analysis of correlation between the TMJ functional and craniofacial morphologic variables and between the TMJ functional and TMJ morphometric variables, considering DS and NDS separately and examining for interside differences. Factor analysis was performed using a Varimax rotation method with Kaiser normalization to extract principal components on each side based on the Scree plot and eigen values. *p* < 0.05 was considered to indicate a statistically significant difference.

## 3. Results

### 3.1. Comparison of Overall Measurements between the Deviated and Nondeviated Sides

The mean values of the 3D craniofacial morphologic (CM) measurements, 3D TMJ morphometric (TM) measurements, and TMJ functional (TF) measurements were compared between the DS and NDS (Table 2). Of the CM variables related to facial asymmetry, the maxillary height and mandibular body length showed no interside difference. In contrast, ramal height was significantly shorter and both frontal and lateral ramal inclinations were greater on the DS than on the NDS. This indicated that the facial asymmetries in our samples were of mandibular asymmetry type with no significant maxillary cant.

Among three categories of TM variable, the extracapsular condylar position relative to the cranial base showed no interside difference. On the other hand, the intracapsular condylar position relative to articular eminence showed interside differences in two variables. Medial joint space and axial condylar angle were significantly larger on the DS than on the NDS. The slope inclination of articular eminence exhibited interside difference only on the anterior wall. Compared with NDS, anterior eminence steepness was significantly greater on the DS.

The TF measurements represented significant bilateral differences except for the opening condylar path length. Compared with the NDS, the protrusive condylar path length, non-working condylar path length, and non-working incisal path length were significantly longer on the DS, while the sagittal condylar inclination during protrusive movement was significantly greater.

### 3.2. Correlation between Craniofacial Morphology, TMJ Morphometry, and TMJ Function

Five CM variables and 10 TM variables were selected for further correlation analysis with five TF variables related to asymmetric condylar movement, with the elimination of three variables related to the extracapsular condylar position that showed no interside differences. A total of 16 variables were finally selected for factor analysis following the additional exclusion of four variables related to posterior and lateral joint spaces and eminence steepness based on the results of correlation analysis. In the present study, factor analysis was performed not to reduce the number of variables for the subsequent correlation analysis but to identify the interrelated variables in the extracted principal components, which resulted in different sets of variables between the DS and NDS.

On the DS, protrusive condylar path length among TF variables showed positive correlation with frontal ramal inclination (*p* < 0.01) out of the CM variables, and positive correlations with anterior joint space (*p* < 0.05), medial joint space (*p* < 0.05), anterior eminence steepness (*p* < 0.01), and axial condylar angle (*p* < 0.05) out of the TM variables (Table 3). As a result of the factor analysis, four principal components were extracted supporting the result of correlation analysis. Variables that were heterogeneously interrelated with all TF variables were categorized into Component 1, and were anterior eminence steepness and frontal ramal inclination (Table 4).

On the NDS, on the other hand, less correlation was found compared with the DS. Of the TF variables, the opening condylar path length had a negative correlation with medial joint space (*p* < 0.001) of the TM variables, while the TF variable of sagittal condylar inclination showed a positive correlation with the TM variable of anterior eminence steepness (*p* < 0.05) (Table 3). Factor analysis resulted in four principal components, and the TF variables were categorized into two components. The TF variables opening condylar path length and sagittal condylar inclination were categorized into Component 1 and showed heterogenous interrelation with the TM variables axial condylar angle and anterior eminence steepness (Table 5).

## 4. Discussion

The present study investigated the asymmetric path of condylar movement in patients with mandibular asymmetry and prognathism using an automated ultrasonic AxioQuick^®^ system matched with CBCT analysis. We exclusively found that the significant morphologic features in craniofacial pattern and TMJ environment correlated with the asymmetric condylar paths during mandibular border movement were different between the DS and NDS, which could be supported by a computerized sensitive real-time jaw-tracking system.

The AxioQuick^®^ recorder system representatively revealed the different patterns of condylar paths between the deviated and non-deviated sides according to the type of mandibular movement. During protrusion, the deviated condyle showed a longer and steeper sagittal path than the non-deviated condyle (Table 2). The deviated condyle showed a medially inclined path, while the non-deviated condyle showed a laterally inclined path, alleviating facial asymmetry. During non-working movement, the deviated condyle exhibited a longer path accompanied by a longer incisor path length, which corresponded to those of previous related studies [3,6,16,26], supporting the idea that the condylar paths tend to compensate for morphologic asymmetry during protrusive and lateral excursion movements. During maximum opening and closing movement, in contrast, no interside differences of condylar paths were observed (Table 2). This was consistent with a study suggesting that lateral mandibular shift was maintained during symmetrical condylar movement such as maximum opening and closing, because the shifted condyle compromises the integrity and synchronism of the condyle-disc assembly in patients with healthy TMJs [4].

With regard to the relationship between asymmetric condylar movements and craniofacial asymmetry, it has been previously reported that vertical and transverse skeletal asymmetries were closely associated with condylar functional asymmetries [27,28]. Hashimoto et al. [6] found that the degree of chin deviation was correlated with frontal maxillary and mandibular plane angles and the right–left difference of mandibular length morphologically, which was correlated with interside differences of condylar paths in all functional movements. Ikeda et al. [23] insisted that the more the morphologic mandibular asymmetry increased, the more the condyles moved to the DS during protrusive movement. The present study assessed greater number of parameters encompassing craniofacial and temporomandibular anatomies as well as condylar dynamics than the previous studies. As a result, however, the craniofacial contributing factor to the asymmetric condylar paths was confined to mandibular asymmetry (Table 3): uniquely the protrusive condylar path length (PCPL) was positively correlated with frontal ramal inclination (FRI) on the DS (Table 4). Considering that the asymmetric functional loading due to vertical skeletal asymmetry—such as maxillary cant with differential occlusal plane steepness between the sides—might be a causative factor of TMD [29], further study is anticipated to compare the condylar movements according to the subtype of facial asymmetry including bimaxillary rolling, yawing and/or translational asymmetry.

For the evaluation of the relationship between asymmetric condylar movements and TMJ morphologic asymmetry, the present study examined the TMJ environmental factors dividing into condyles, joint spaces, and articular eminences. Previous studies have investigated laterality in the TMJ space in subjects with skeletal asymmetry [6,16,19,30]. The recent consensus is that the deviated condyle tends to be located more superiorly and posteriorly and rotated more medially [30], and shows a longer and steeper path during protrusion and lateral excursion [16]. This could be explained by steeper anterior articular eminence, as compensatory responses [6,19]. As a result of our study examining two relative positions of condyles, condylar positional asymmetry was marked by differential medial joint space (MJS) and axial condylar angle (ACA) within the glenoid fossa, showing no differences when evaluated relative to the cranial base (Table 2). Furthermore, MJS and ACA were positively correlated with a PCPL on the DS, though only the MJS was negatively correlated with opening condylar path length (OCPL) on the NDS (Table 3). In addition, the steeper the anterior wall of articular eminence (AES), the more the PCPL increased on the DS, and the greater the increase in sagittal condylar inclination (SCI) on the NDS. Taken together, asymmetric condylar movements between the DS and NDS in facial asymmetry patients were closely correlated with the TMJ morphologic asymmetry, rather than with craniofacial asymmetry. This might be the reason why the asymmetric condylar path length remained unchanged even after orthognathic surgery for the correction of craniofacial asymmetry [6].

As a result of factor analysis to support the correlations among lots of parameters, different interside relationships between morphological and functional variables could be confirmed. On the DS, all tested TMJ functional variables—NCPL, PCPL, NIPL, SCI, and OCPL—showed significant interrelationship as the first principal component, having close correlation with two morphologic variables of AES and FRI (Table 4). On the NDS, only two functional variables—OCPL and SCI—showed a correlation with morphologic variables of ACA and AES (Table 5). In consistent with the correlation analysis, these findings imply that CBCT morphometric analysis of craniofacial pattern and TMJ anatomy are not enough to exactly predict the pattern of condylar movements. Direct real-time functional analysis on the patterns and limits of condylar paths in 3D using a computerized jaw-tracking system would be very helpful for accurate diagnosis especially in patients with craniofacial deformities.

It should be considered that the condylar position in the glenoid fossa and the condylar movement depend on other environmental elements, including TMJ discs and ligaments [21], masticatory muscles [31], the occlusal scheme [32], and the bony structures of the TMJ and face [33]. The ACA and AES may be genetically determined via asymmetric craniofacial growth [12] and may change due to asymmetrical muscle function environmentally [31], creating an asymmetric sagittal condylar path length and inclination. Nonetheless, the association between the condylar movement and the soft tissues of the stomatognathic system has not been elucidated due to the limitations of quantitative evaluation. Therefore, in our study, we excluded subjects with TMD signs and symptoms and neuromuscular disorders such as trismus. Conversely, the position of the condyles in the fossa might also affect the shape of the glenoid fossa and mandibular asymmetry. Accordingly, the asymmetric condylar position and movements need to be understood as the sum of adaptational responses to the asymmetric soft tissue functions and of compensatory interactions with asymmetric development of the mandible and glenoid fossa [23].

This study has some experimental limitations. Subjects were not categorized according to the sagittal or vertical skeletal patterns. In addition to facial asymmetry, patients presenting skeletal Class III with mandibular prognathism were included without control group having different sagittal skeletal discrepancy like Class I or Class II. Saccucci et al. [34] demonstrated that the condylar volume may differ between the DS and NDS in patients with mandibular asymmetry based on the fact that skeletal Class III patients had significantly greater condylar volume than Class II subjects. Hashimoto et al. [6] found that the condylar unit length and unit volume were significantly smaller on the DS than on the NDS in patients with mandibular asymmetry, which could not be verified in the present study. Rather, to rule out the possible effects of different anatomical structures of the glenoid fossa and condylar position among different sagittal skeletal patterns [21,35], we intended to specify the sample characteristics as facial asymmetry with skeletal Class III with mandibular prognathism and healthy TMJs. Lastly, the AxioQuick^®^ recorder system has a fundamental weak point of desensitization of the sensors in patients with severe mandibular asymmetry. This is because the distance between the transmitters inducing ultrasound pulses on the mandibular part and the receivers on the head part may increase beyond the critical distance range. Development of a modified tracking device is demanding when it comes to compensating the increased inter-sensors distance or to increasing the pulse transmitting capacity in patients with severe deformity.

Further study is anticipated to compare the correlation patterns in patients with various subtypes of craniofacial deformities, with or without TMD problems. Moreover, with more advanced digitized dynamic analysis techniques like electromyography and digitized occlusal analysis as well as this jaw-tracking system, more updated information including the roles of facial and TMJ soft tissues could be drawn in an integrated manner.

## 5. Conclusions

A computerized and automated real-time ultrasonic jaw tracking system representatively revealed the different patterns of condylar paths between the DS and NDS during protrusive and lateral mandibular movements in patients with facial asymmetry and mandibular prognathism. Although TMJ morphometric variables like AES or ACA showed significant correlations with mandibular movement variables, we could not predict every condylar path from CBCT morphometric variables. More advanced techniques for orofacial dynamic analysis are anticipated to reach an integrated clinical relevance on the craniofacial morphology and functions in patients with severe craniofacial deformity and functional problems.

## Figures and Tables

**Figure 1 sensors-21-02591-f001:**
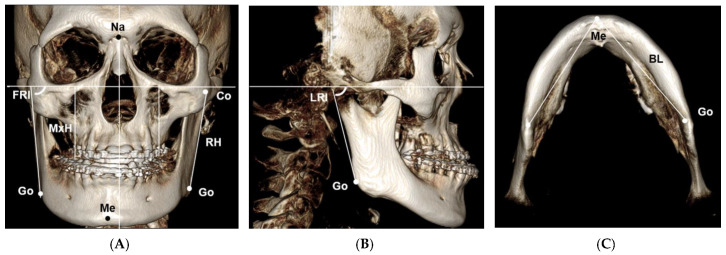
(**A**–**C**) Landmarks and parameters for the measurements of craniofacial morphology. Na: nasion; Me: menton; Go: gonion; Co: condyle; MxH: maxillary height; RH: ramal height; FRI: frontal ramal inclination; LRI: lateral ramal inclination; BL: mandibular body length.

**Figure 2 sensors-21-02591-f002:**
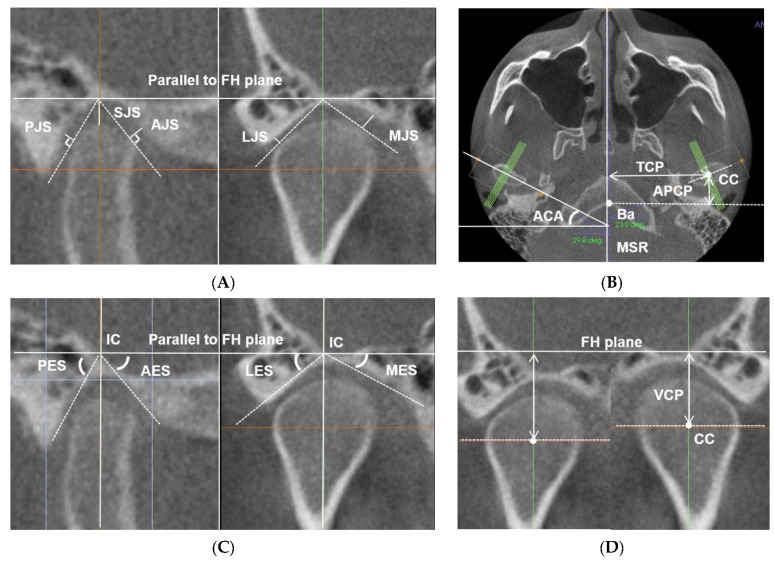
(**A**–**D**) Landmarks and parameters for the measurements of temporomandibular joint (TMJ) morphometric variables. Ba: basion; CC: center of the condyle; AJS: anterior joint space; SJS: superior joint space; PJS: posterior joint space; LJS: lateral joint space; MJS: medial joint space: ACA: axial condylar angle; APCP: anteroposterior condylar posture; TCP: transverse condylar posture; VCP: vertical condylar posture; AES: anterior eminence steepness; PES: posterior eminence steepness; LES: lateral eminence steepness; MES: medial eminence steepness; IC: intersecting point.

**Figure 3 sensors-21-02591-f003:**
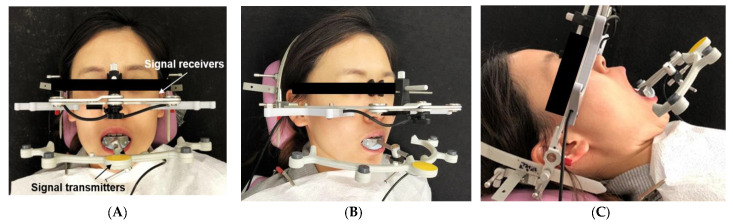
(**A**–**C**) Recording of jaw movements using the AxioQuick^®^ recorder. There were 8 ultrasonic signal receivers embedded in the head frame and 4 signal transmitters embedded on a face bow. Face bow was attached to mandibular incisors using a dental occlusal clutch. The clutch allowed the patient’s mandible to move freely. (**A**): Frontal view. (**B**): Oblique view. (**C**): Maximum opening movement in an inclined supine position.

**Figure 4 sensors-21-02591-f004:**
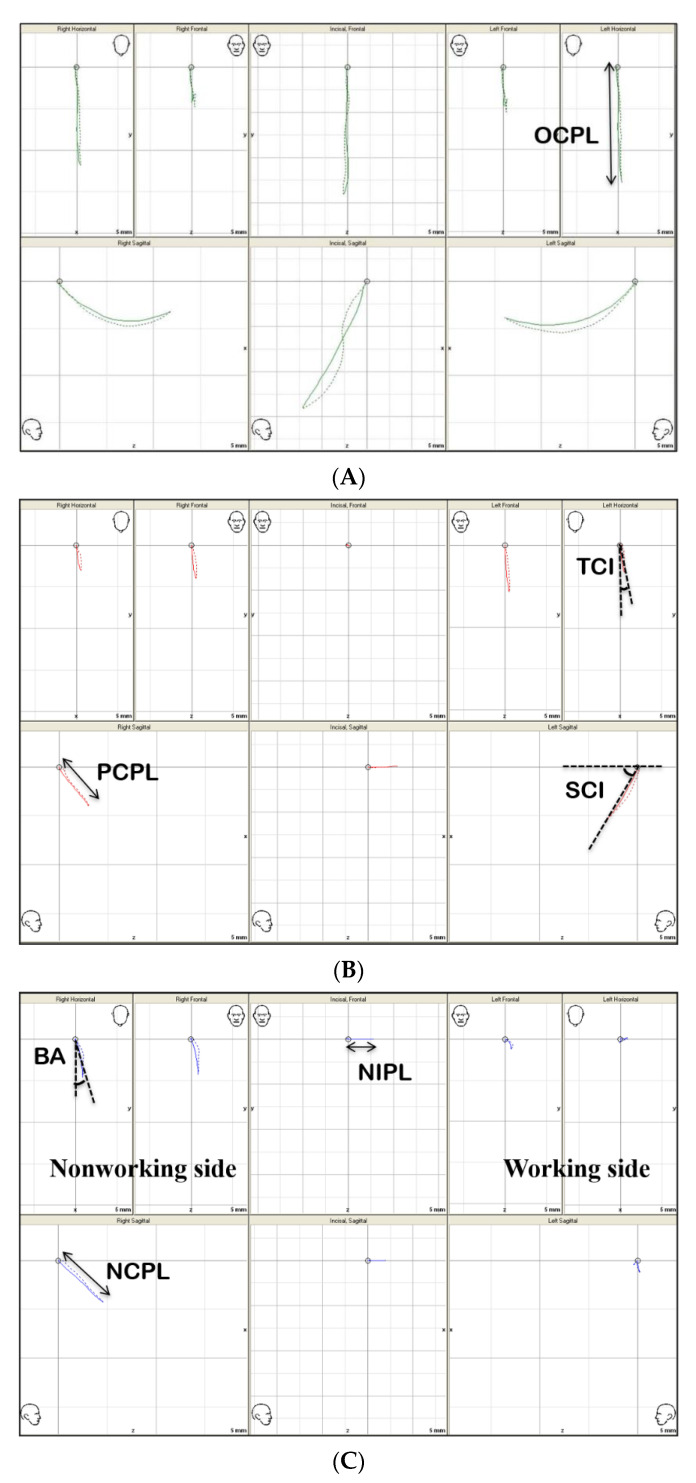
(**A**–**C**) The computer-assisted measurement and mapping software program for geometric analysis of jaw movements. (**A**): maximum open-close. (**B**): maximum protrusion. (**C**): maximum lateral excursion. OCPL: opening condylar path length; PCPL: protrusive condylar path length; SCI: sagittal condylar inclination during protrusive movement; TCI: transverse condylar inclination; NIPL: lateral incisal path length; NCPL: lateral non-working condylar path length at balancing side; BA: Bennett angle.

**Table 1 sensors-21-02591-t001:** Sample size analysis for 90% power, with an alpha value of 0.05.

Representative Variables	Deviation Side (*n* = 30)	Non-Deviation Side (*n* = 30)	*p*-Value	Effect Size	Power (%)	No. of 80% Power (Each Group)
Anterior eminence steepness (AES) [°]	46.54 ± 10.97	39.87 ± 8.66	<0.000 ***	0.6763	93.17	20
Protrusive condylar path length (PCPL) [mm]	6.97 ± 2.09	6.12 ± 1.83	0.001 **	1.0164	99.94	10
Non-working condylar path length (NCPL) [mm]	7.40 ± 1.98	6.02 ± 1.78	<0.000 ***	0.9225	99.69	12

* *p* < 0.05; ** *p* < 0.01; *** *p* < 0.001.

**Table 2 sensors-21-02591-t002:** Comparison of craniofacial morphologic, TMJ morphometric, and TMJ functional variables between deviated and non-deviated sides.

Compartment		Variables	Deviated Side		Non-Deviated Side		*p*-Value
			Mean	SD	Mean	SD	
Craniofacial morphology (cm)		Maxillary height (MxH) [mm]	49.67	3.56	50.45	3.30	0.264
		Ramal height (RH) [mm]	45.91	5.39	48.40	4.24	0.028 *
		Frontal ramal inclination (FRI) [°]	82.34	3.70	77.00	3.31	0.000 ***
		Lateral ramal inclination (LRI) [°]	86.41	6.84	83.93	5.59	0.014 *
		Mandibular body length (BL) [mm]	74.62	5.05	75.58	5.72	0.129
TMJ morphometry (tm)	Condyle position relative to cranial base	Anteroposterior condylar posture (APCP) [mm]	13.21	3.33	13.80	3.16	0.234
		Transverse condylar posture (TCP) [mm]	52.03	2.51	52.77	2.40	0.137
		Vertical condylar posture (VCP) [mm]	8.72	3.07	8.29	3.05	0.275
	Condyle position relative to eminence	Anterior joint space (AJS) [mm]	2.13	0.71	1.74	0.53	0.362
		Superior joint space (SJS) [mm]	2.05	0.71	2.26	0.93	0.516
		Posterior joint space (PJS) [mm]	1.59	0.46	2.01	0.68	0.882
		Medial joint space (MJS) [mm]	2.65	0.69	1.60	0.51	0.002 **
		Lateral joint space (LJS) [mm]	1.48	0.59	1.77	0.49	0.377
		Axial condylar angle (ACA) [°]	21.18	6.37	17.23	6.00	0.019 *
	Eminence steepness	Anterior eminence steepness (AES) [°]	46.49	9.21	39.10	7.39	0.000 ***
		Posterior eminence steepness (PES) [°]	58.30	9.03	58.84	6.60	0.988
		Medial eminence steepness (MES) [°]	48.72	12.17	49.10	9.29	0.802
		Lateral eminence steepness (LES) [°]	37.77	7.97	39.62	6.73	0.066
TMJ function (tf)		Opening condylar path length (OCPL) [mm]	15.27	2.75	14.21	3.60	0.338
		Protrusive condylar path length (PCPL) [mm]	9.18	1.73	6.88	1.64	0.001 **
		Sagittal condylar inclination (SCI) [°]	44.39	5.21	39.57	7.48	0.002 **
		Transverse condylar inclination (TCI) [°]	4.20	2.86	−2.44	2.23	0.031 *
		Bennet angle (BA) [°]	14.39	3.99	9.11	2.35	0.041 *
		Non-working condylar path length (NCPL) [mm]	8.15	1.46	6.12	1.70	0.000 ***
		Non-working incisal path length (NIPL) [mm]	6.59	1.18	5.81	1.03	0.001 **

Analyzed by paired *t*-test: * *p* < 0.05; ** *p* < 0.01; *** *p* < 0.001.

**Table 3 sensors-21-02591-t003:** Correlation between TMJ functional parameters (tf) and craniofacial morphologic (cm) and TMJ morphometric (tm) parameters.

			Deviated Side					Non-Deviated Side				
TMJ Function (tf)			OCPL [mm]	PCPL [mm]	SCI [°]	NIPL [mm]	NCPL [mm]	OCPL [mm]	PCPL [mm]	SCI [°]	NIPL [mm]	NCPL [mm]
**Cranio** **facial morphology [cm]**	MxH [mm]	R	0.137	−0.168	0.190	−0.067	0.252	0.101	−0.025	0.203	0.204	0.080
		*P*	0.488	0.392	0.334	0.734	0.196	0.609	0.898	0.301	0.297	0.685
	RH [mm]	R	0.116	−0.157	−0.155	−0.168	−0.092	0.094	0.130	−0.049	−0.234	0.222
		*P*	0.558	0.424	0.431	0.393	0.642	0.636	0.509	0.803	0.231	0.257
	FRI [°]	R	0.257	0.787	0.245	0.242	0.209	−0.282	−0.284	0.153	0.095	−0.051
		*P*	0.186	0.009 **	0.209	0.214	0.286	0.146	0.144	0.438	0.632	0.795
	LRI [°]	R	0.134	0.025	0.302	0.139	0.075	0.221	0.231	0.332	0.041	0.025
		*P*	0.498	0.900	0.119	0.480	0.705	0.259	0.238	0.084	0.834	0.900
	BL [mm]	R	−0.091	−0.170	0.035	−0.352	−0.096	−0.270	−0.001	0.025	0.012	0.067
		*P*	0.646	0.387	0.860	0.067	0.627	0.164	0.998	0.899	0.953	0.736
**TMJ** **morphometry** **[tm]**	AJS [mm]	R	−0.017	0.612	0.046	0.270	0.316	0.168	0.176	0.237	−0.236	−0.189
		*P*	0.930	0.029 *	0.817	0.165	0.102	0.393	0.370	0.225	0.227	0.335
	SJS [mm]	R	−0.019	0.189	0.157	0.212	0.117	0.348	0.203	−0.109	0.127	0.045
		*P*	0.924	0.336	0.424	0.278	0.553	0.070	0.299	0.581	0.520	0.819
	PJS [mm]	R	−0.219	−0.004	0.032	0.165	−0.002	0.360	0.183	−0.138	0.233	0.000
		*P*	0.263	0.984	0.872	0.402	0.992	0.060	0.351	0.485	0.233	0.998
	MJS [mm]	R	−0.008	0.675	−0.020	0.347	0.210	−0.845	0.321	−0.239	0.167	0.105
		*P*	0.966	0.011*	0.918	0.070	0.284	0.000 ***	0.096	0.221	0.396	0.595
	LJS [mm]	R	0.174	0.336	0.242	0.327	0.150	0.451	0.255	0.169	0.335	0.166
		*P*	0.376	0.081	0.215	0.089	0.446	0.105	0.190	0.390	0.081	0.397
	ACA [°]	R	0.183	0.713	0.046	0.337	0.194	0.267	0.334	0.239	−0.102	−0.174
		*P*	0.352	0.029 *	0.816	0.079	0.323	0.170	0.082	0.221	0.604	0.377
	AES [°]	R	0.264	0.797	0.348	0.595	0.335	0.191	0.164	0.736	0.151	0.204
		*P*	0.175	0.007 **	0.069	0.105	0.081	0.329	0.403	0.020*	0.442	0.299
	PES [°]	R	−0.144	−0.044	0.284	−0.147	0.082	−0.240	−0.100	0.233	0.173	0.283
		*P*	0.466	0.825	0.143	0.456	0.678	0.218	0.614	0.232	0.379	0.145
	MES [°]	R	0.332	−0.127	0.076	−0.215	0.134	−0.106	−0.126	−0.172	0.087	−0.006
		*P*	0.084	0.518	0.702	0.271	0.498	0.591	0.521	0.380	0.659	0.978
	LES [°]	R	−0.056	−0.057	0.003	0.005	−0.045	0.129	−0.028	0.071	0.180	0.103
		*P*	0.776	0.772	0.988	0.978	0.821	0.512	0.889	0.720	0.360	0.601

* *p* < 0.05; ** *p* < 0.01; *** *p* < 0.001.

**Table 4 sensors-21-02591-t004:** Factor analysis with extraction of 4 principal components as linear combinations of the original 16 variables on the deviation side.

Variables	Components			
	1	2	3	4
NCPL_tf	0.773	−0.061	−0.056	−0.059
PCPL_tf	0.765	0.213	−0.036	−0.371
AES_tm	0.690	0.502	−0.032	0.080
NIPL_tf	0.644	0.288	−0.224	−0.263
FRI_cm	0.634	0.044	0.194	−0.016
SCI_tf	0.530	−0.108	−0.216	0.365
OCPL_tf	0.449	−0.112	−0.063	0.198
MES_tm	0.187	−0.751	0.146	0.059
MJS_tm	0.407	0.676	0.438	−0.124
SJS_tm	0.183	0.659	0.133	0.328
ACA_tm	0.283	0.651	0.038	−0.250
LRI_cm	0.119	0.272	−0.804	0.313
BL_cm	−0.074	−0.028	0.759	0.157
RH_cm	−0.040	0.151	0.706	0.391
MxH_cm	0.154	−0.011	0.308	0.722
AJS_tm	0.320	0.207	0.049	−0.708

Extraction method: principal component analysis; rotation method: Varimax with Kaiser normalization.

**Table 5 sensors-21-02591-t005:** Factor analysis with extraction of 4 principal components as linear combinations of the original 16 variables on the non-deviated side.

Variables	Components			
	1	2	3	4
ACA_tm	0.769	−0.206	−0.047	0.079
OCPL_tf	0.729	0.063	−0.492	0.311
SCI_tf	0.680	0.360	0.461	−0.036
AES_tm	0.670	0.349	0.204	−0.085
NCPL_tf	−0.037	0.865	0.005	0.066
NIPL_tf	0.017	0.804	−0.017	−0.014
PCPL_tf	0.381	0.638	−0.353	0.158
FRI_cm	0.038	−0.014	0.741	−0.045
MJS_tm	0.027	0.242	0.706	−0.173
LRI_cm	0.013	0.108	−0.671	−0.330
AJS_tm	0.270	−0.326	0.066	0.702
SJS_tm	0.190	0.152	−0.278	0.700
RH_cm	−0.083	0.119	0.011	0.587
BL_cm	−0.158	0.236	0.216	−0.551
MES_tm	−0.198	−0.096	0.295	0.536
MxH_cm	−0.137	0.290	0.120	0.360

Extraction method: principal component analysis; rotation method: Varimax with Kaiser normalization.

## Data Availability

Not applicable.

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
