# Peer review of "Automated Real-Time Evaluation of Condylar Movement in Relation to Three-Dimensional Craniofacial and Temporomandibular Morphometry in Patients with Facial Asymmetry"

_sensors, 2021, doi:10.3390/s21082591_

Round 1

Reviewer 1 Report

I cannot find anything wrong in this paper. Excellent work. Although not really close to my field of expertise, I found it easy to follow and understand. The use of English is perfect. 

Author Response

Manuscript ID: sensors-1120117   

Title: Automated Real-Time Evaluation of Condylar Movement in Relation to Three-Dimensional Craniofacial and Temporomandibular Morphometry in Patients with Facial Asymmetry

Dear editors and reviewers,

We would like to appreciate editors and reviewers for giving us the opportunity of revision with elaborate comments. We’ve learned a lot from reviewers’ instructive comments and recommendations. We tried to address all the concerns raised by reviewers faithfully. We hope that you could make a favorable arrangement for our revision even though some limitations might remain inevitably in the manuscript. Your consideration is highly appreciated by all authors. Thank you very much again.

-------------------------------------------------------------------------------------------------------------

Response to Reviewer 1 Comments

Point 1: I cannot find anything wrong in this paper. Excellent work. Although not really close to my field of expertise, I found it easy to follow and understand. The use of English is perfect.

Response 1: We really appreciate for your generous revision with compliments. Your kind approval of our work would be a great driving force for us to contribute to further related research. Thank you very much.

Reviewer 2 Report

This is a very well presented article that will have relevance especially to clinically oriented readers. For technical readers, more detail and emphasis on the experimental apparatus may be required.

It appears that the authors are using a piece of off-the-shelf hardware that has not been validated for the application. The authors should cite or describe any studies that indicate that the device performs as intended in the environment of use, otherwise they cannot know that the device is making correct measurements especially in the case of deformed anatomy. For example, could the device itself be affecting the measurements, since it may have been designed for different anatomy?

Similarly the authors rely on an off-the-shelf software system. While this package is well known, some basic validation, testing or citations that indicate that the system is being used correctly would be helpful. Validation might consist of a few measurements performed on a test phantom for example.

A few specific comments:

Line 167: Can the authors indicate the US receivers and transmitters with arrows on some of them in the photo.

Line 173: US motion tracking systems are known to produce variable results depending on ambient temperature. Can the authors say anything about room temperature and ambient noise during the studies.

Line 178: Frequency of sample collection? Total samples collected? Duration of each motion collected? How were the samples collected? Was the subject instructed to perform a particular motion, was the motion practiced before the data was collected, was the jaw manipulated manually etc.

Line 190: “No repeated measurement”. Some evaluation and discussion on repeatability of the movement studies are required and how the authors decided that it was not necessary to repeat or average measurements. Perhaps this has been done by others but this should be cited. This validation could be conducted within a single patient is required even if it is a reference to another paper, but it is preferred that at least some repeat measurements be performed on one of your subjects to validate.

Were the extents of the deformities enough to make repeated measurements substantially different for the same patient? Should the repeated measurements be averaged together?

Some acronyms are used only once or twice e.g. BA, SJS etc.  or defined both in the text and the figure caption. In fact there are a lot of abbreviations in this paper that make it a little difficult to read for technical readers unfamiliar with  maxillofacial anatomy. Perhaps repeat the long form on a few infrequently used acronyms.

Author Response

Manuscript ID: sensors-1120117   

Title: Automated Real-Time Evaluation of Condylar Movement in Relation to Three-Dimensional Craniofacial and Temporomandibular Morphometry in Patients with Facial Asymmetry

Dear editors and reviewers,

We would like to appreciate editors and reviewers for giving us the opportunity of revision with elaborate comments. We’ve learned a lot from reviewers’ instructive comments and recommendations. We tried to address all the concerns raised by reviewers faithfully. We hope that you could make a favorable arrangement for our revision even though some limitations might remain inevitably in the manuscript. Your consideration is highly appreciated by all authors. Thank you very much again.

-------------------------------------------------------------------------------------------------------------

Response to Reviewer 2 Comments

Point 1: This is a very well presented article that will have relevance especially to clinically oriented readers. For technical readers, more detail and emphasis on the experimental apparatus may be required.

Response 1: Your kind consideration and is highly appreciated. We totally agree that the technical points in our manuscript is relatively weak. To be faithful to your valuable comments, we added specific descriptions on the technical principle and experimental apparatus of Axioquick recorder. Changed parts in the manuscript are remarked by red texts. We hope that you could make a favorable arrangement for our revision even though some limitations might remain inevitably.

Point 2: It appears that the authors are using a piece of off-the-shelf hardware that has not been validated for the application. The authors should cite or describe any studies that indicate that the device performs as intended in the environment of use, otherwise they cannot know that the device is making correct measurements especially in the case of deformed anatomy. For example, could the device itself be affecting the measurements, since it may have been designed for different anatomy?

Response 2: Thank you for your instructive comments. To convince the readers of the reliability and validity of our jaw-tracking assessment system, we described more in details about its significant performance with citation of more references in the introduction section, as below;

(line 45, page 2) As a computerized ultrasonic axiography, the AxioQuick® recorder (SAM Co., Munich, Germany is specialized in quantitative analysis of the direction and amount of condylar paths within the glenoid fossa during mandibular border movement [13-15]. This registration system is based on the measurement of real-time latency periods of sequentially transmitted ultrasound pulses between four transmitters attached to the mandible and eight receivers mounted on a face bow [14]. The improved resolution of the 3D sensors increased the signal quality and diagnostic validity. Inclusion of lightweight tiny sensors could enhance the patient’s comfort during jaw movement and thus decrease the measurement errors. High diagnostic specificity and sensitivity of this system has been proven in both children [13] and adults with healthy or pathologic TMJs [14,15]. Based on the reliability and validity on the clinical relevance, a superior AxioQuick® recorder system was introduced in the present study to investigate the diagnostic value of this to analyze various condylar movements in the specific condition of dentofacial asymmetry.

In addition, as you concerned, there has been little studies investigating the condylar movement using this ultrasonic computerized recorder system in patients with various craniofacial deformities. Actually for that reason, we had designed this study to evaluate the performance accuracy of AxioQuick recorder system in anatomically specific TMJ condition, and facial asymmetry was firstly involved in our samples that might have asymmetric TMJ environment. Further we aimed to find out the relationship between the anatomic variations and real functional movements. We hope you could consider this point as one significance of our paper.

Point 3: Similarly the authors rely on an off-the-shelf software system. While this package is well known, some basic validation, testing or citations that indicate that the system is being used correctly would be helpful. Validation might consist of a few measurements performed on a test phantom for example.

Response 3: Following your valuable recommendation, we cited more references indicating both hardware and software characteristics on the basis of basic validation using critical parameters, as we explained to the Point 2 (above). The most frequent parameters that were used for the validation test was horizontal condylar path inclination angle and linear distance. Considering that the previously tested parameters have not been enough, we exclusively included 7 test parameters representing the movements in all dimensions.

The AxioQuick recorder system consists of full package of device sets and compatible computer software program, since it is a kind of computerized real-time screening system. This software generates the real-time digital data and real-time analysis automatically based on the 3D coordinated system. We added the description on the digital software in the method section as below;

(line 181, page 6) Axis-horizontal plane was established as a zero-reference plane for all 3D mandibular movements. The patient’s terminal hinge axis was determined as the condylar reference point by rotational analysis in the software program automatically. The condylar paths were recorded during the mandibular border movement in three dimensions: maximum opening and closing, protrusion, and working and nonworking lateral movements. Along with 3D acquisition of movement, the software generated a real-time digital data. The X, Y, and Z axes-based coordinated information was obtained for automatic quantification.

A few specific comments:

Point 4: Line 167: Can the authors indicate the US receivers and transmitters with arrows on some of them in the photo.

Response 4: We added the indication arrows on Figure 3-(a) as you mentioned.

Point 5: Line 173: US motion tracking systems are known to produce variable results depending on ambient temperature. Can the authors say anything about room temperature and ambient noise during the studies.

Response 5: We appreciate for your considerate comments. We measured each patient in an isolated room equipped with this system, where room temperature is maintained without noise. We tried to eliminate any environmental factors which might affect the patient’s condition and tension throughout the testing period. We added this remark in the method section as below;

(line 171, page 5) All measurements were performed in an isolated room equipped with this system where the room-temperature is maintained without noise.

Point 6: Line 178: Frequency of sample collection? Total samples collected? Duration of each motion collected? How were the samples collected? Was the subject instructed to perform a particular motion, was the motion practiced before the data was collected, was the jaw manipulated manually etc.

Response 6: We apologize for missing the description on the sample selection. We retrospectively screened 43 patients with facial asymmetry and mandibular prognathism who were tested using AxioQuick recorder before orthodontic treatment, from January 2017 to December 2019. Based on our inclusion and exclusion criteria, finally analyzed sample was 30. Following your important comments, we explained the details about sample recruitment and testing procedure in the material and method section, as below;

(line 92, page 2) Forty three patients with facial asymmetry and mandibular prognathism, who had been tested using an ultrasonic jaw-tracking system before orthodontic treatment, from January 2017 to December 2019.

(lines 173, page 5) Each patient was seated in an inclined supine position with the full device sets on the head and the mandible. Each patient was instructed to practice and repeat all tested mandibular movements until representative real-time estimates of all dynamic parameters could be obtained.

Point 7: Line 190: “No repeated measurement”. Some evaluation and discussion on repeatability of the movement studies are required and how the authors decided that it was not necessary to repeat or average measurements. Perhaps this has been done by others but this should be cited. This validation could be conducted within a single patient is required even if it is a reference to another paper, but it is preferred that at least some repeat measurements be performed on one of your subjects to validate.

Response 7: We are sorry for the confusion in the description. We agree that repeated measurements and averaged values are necessary to validate the intraexaminer reliability in this kind of experiment. Of course, each patient was trained by an expert to repeat the mandibular border movements in 3D until the representative estimates could be obtained. This electronic recording system allowed the examiner to confirm in real time if the patient could reach the reproducible movements and turn-back to the calibrated starting position, representing geometrically stable patterns. Most of patients got there in 2~3 movement cycles on average, and got more relaxed. At that trained point, we took two measurement data sets (actually showing negligible numeric difference between the two sets in the real-time report) and chose the averaged value of each parameter for the analysis. We corrected the corresponding part as below;

(line 174, page 5) Each patient was instructed to practice and repeat all tested mandibular movements until representative real-time estimates of all dynamic parameters could be obtained. At that time point, two measurement data sets were acquired in each patient. The averaged value of each parameter was taken for the analysis.

Point 8: Were the extents of the deformities enough to make repeated measurements substantially different for the same patient? Should the repeated measurements be averaged together?

Response 8: We really appreciate for your considerate concerns. The 3D CBCT analysis was performed not only to reveal the extent and the pattern of craniofacial deformity in patients with facial asymmetry with mandibular prognathism, but also to find any temporomandibular morphological or positional discrepancy which might be significantly related to the mandibular movements. For both of them, the computerized software-based repeated measurements were performed by one experienced examiner in a 2-week interval, and we took the averaged data sets for further statistical analysis, based on the high level of intra-examiner reliability as indicated by Dahlberg’s formula. As we described in the method section, the measurement errors ranged from 0.02 to 0.16 mm for the linear parameters and from 0.04 to 0.31 degrees for the angular parameters in the CBCT analysis. We added the description about this to be more clarified as below;

(Line 108, Page 3) One experienced examiner (LWJ) performed the CBCT image reconstruction and measurements two time in a 2 week-interval. The averaged values of the two data sets was taken for the analysis. The method error for each parameter was calculated using Dahlberg’s formula. The measurement errors ranged from 0.02 to 0.16 mm for the linear parameters and from 0.04 to 0.31 degrees for the angular parameters, indicating high intraexaminer reliability. 3D morphometric analysis was conducted in two aspects: craniofacial and TMJ morphologies.

Point 9: Some acronyms are used only once or twice e.g. BA, SJS etc.  or defined both in the text and the figure caption. In fact there are a lot of abbreviations in this paper that make it a little difficult to read for technical readers unfamiliar with maxillofacial anatomy. Perhaps repeat the long form on a few infrequently used acronyms.

Response 9: We totally understand that lots of unfamiliar acronyms to describe the reference points for the morphometric analysis will make it difficult for the technical readers to read this paper. We appreciate for your kind suggestion like this, considering that this reference points are indispensable to interpret the results of morphometric analysis in relation to functional parameters from jaw-tracking system. We tried to minimize the acronyms by repeating the full term for the infrequently used ones.

Reviewer 3 Report

This paper is not well written.

Results should be discussed in clear way.
For example "Taken together, it can be postulated that TMJ morphometric asymmetry more than
the craniofacial morphologic asymmetry seems to be reflected in the functional asym-
metry, representing different correlations in between the DS and NDS, as supported by
factor analysis."

this only one place in Discussion where PCA or factor analysis (Tab.5) are mentioned.
Tab. 4 nad 5 are not well referenced in Discussion. References should be direct "obtained value XYZ from Tab.4/5 ...is/do/... interesting becasue".
There is lack of direct comparisons between values.

Conclusions are weak also:
"Automated real-time tracking system..." - this is not a good conclusion, because this paper is weak contribution in Sensors area.
This is good paper for any medical journal, but technical aspects are not considered in this paper.
There are some measurements, using commercial system, without any technical contributions of authors.

The Discussion section is very weakly related to the Result section. 
Results from Tab.2-5 are not considered in the Discussion section.

mm3 <- '3' should be the upper index

l.210
"<.05" <- "<0.05"

Author Response

Manuscript ID: sensors-1120117   

Title: Automated Real-Time Evaluation of Condylar Movement in Relation to Three-Dimensional Craniofacial and Temporomandibular Morphometry in Patients with Facial Asymmetry

Dear editors and reviewers,

We would like to appreciate editors and reviewers for giving us the opportunity of revision with elaborate comments. We’ve learned a lot from reviewers’ instructive comments and recommendations. We tried to address all the concerns raised by reviewers faithfully. We hope that you could make a favorable arrangement for our revision even though some limitations might remain inevitably in the manuscript. Your consideration is highly appreciated by all authors. Thank you very much again.

Response to Reviewer 3 Comments

Point 1: This paper is not well written.

Response 1: We thankfully got invaluable feedbacks from your considerate concerns. We really appreciate for giving us one more opportunity of revision despite the lack of logical organization and scientific description. We tried to address all the concerns raised by reviewers following your comments faithfully. We hope that you could make a favorable arrangement for our revision even though some limitations might remain inevitably in the manuscript. Thank you very much again.

Point 2: Results should be discussed in clear way. For example "Taken together, it can be postulated that TMJ morphometric asymmetry more than the craniofacial morphologic asymmetry seems to be reflected in the functional asymmetry, representing different correlations in between the DS and NDS, as supported by factor analysis."

Response 2: We really appreciate for your fundamental comments. We apologize for the confusion with unclear description especially in the discussion section. We attempted to integrate the study results into more practical messages, however I failed to deliver them clearly. We changed the obscure expressions here and there in the discussion section to be more clarified and more relevant to our results, as you indicated. We corrected the above sentence as below;

(Line 346, Page 13) Taken together, asymmetric condylar movements between the DS and NDS in facial asymmetry patients were closely correlated with the TMJ morphologic asymmetry, rather than with craniofacial asymmetry.

Point 3: this only one place in Discussion where PCA or factor analysis (Tab.5) are mentioned. Tab. 4 and 5 are not well referenced in Discussion. References should be direct "obtained value XYZ from Tab.4/5 ...is/do/... interesting because". There is lack of direct comparisons between values.

Response 3: Thank you for your meticulous indication. So sorry for missing that point. We explained the interpretation and significance of the factor analysis by adding a paragraph in the discussion section, as below;

(Line 351, Page 13) As a result of factor analysis to support the correlations among lots of parameters, different interside relationships between morphological and functional variables could be confirmed. On the DS, all tested TMJ functional variables –NCPL, PCPL, NIPL, SCI, and OCPL- showed significant interrelationship as the first principal component, having close correlation with two morphologic variables of AES and FRI (Table 4). On the NDS, only two functional variables – OCPL and SCI- showed correlation with morphologic variables of ACA and AES (Table 5). In consistent with the correlation analysis, these findings imply that CBCT morphometric analysis of craniofacial pattern and TMJ anatomy are not enough to exactly predict the pattern of condylar movements. Direct real-time functional analysis on the patterns and limits of condylar paths in 3D using a computerized jaw-tracking system would be highly helpful for accurate diagnosis especially in patients with craniofacial deformities.  

Point 4: Conclusions are weak also: "Automated real-time tracking system..." - this is not a good conclusion, because this paper is weak contribution in Sensors area. This is good paper for any medical journal, but technical aspects are not considered in this paper. There are some measurements, using commercial system, without any technical contributions of authors.

Response 4: Thank you for your instructive comments. We made a new conclusion dealing with the significances in both clinical and technical aspects, as below. We hope that you could accept this revised conclusion even though the technical contribution of this paper may be still weak.

(Line 403, Page 14) A computerized and automated real-time ultrasonic jaw tracking system representatively revealed the different patterns of condylar paths between the DS and NDS during protrusive and lateral mandibular movements in patients with facial asymmetry and mandibular prognathism. Although TMJ morphometric variables like AES or ACA showed significant correlations with mandibular movement variables, we could not predict every condylar path from CBCT morphometric variables. More advanced techniques for orofacial dynamic analysis are anticipated to reach an integrated clinical relevance on the craniofacial morphology and functions in patients with severe craniofacial deformity and functional problems.

Point 5: The Discussion section is very weakly related to the Result section. Results from Tab.2-5 are not considered in the Discussion section.

Response 5: We appreciate your fundamental indication. We comprehensively revised the discussion section with proper citation of each table and references in there. We added the paragraph or sentences describing the technical implication of our study as well as clinical relevance, to be logically related to the results of statistical analysis. Please review the overall discussion section revised. It would be highly appreciated if you could be generous with our insufficient endeavors, and favorably arrange this revision. Based on this, we will try to go further for subsequent studies following your instructive comments and kind recommendation. Thank you very much again.

Point 6: mm3 <- '3' should be the upper index & l.210 "<.05" <- "<0.05"

Response 6: We really appreciate for your meticulous comments. We corrected all of them as you mentioned. Thank you.

Round 2

Reviewer 3 Report

ok